# Enhancing vision-language models for medical imaging: bridging the 3D gap with innovative slice selection

**Yuli Wang**[1], **Jian Peng**[2], **Yuwei Dai**[1], **Craig Jones**[1], **Haris Sair**[1], **Jinglai Shen**[3],
**Nicolas Loizou**[1], **Jing Wu**[2], **Wen-Chi Hsu**[1], **Maliha Imami**[1], **Zhicheng Jiao**[4],
**Paul Zhang**[5], **Harrison Bai**[1]
[1]Johns Hopkins University
[2]Second Xiangya Hospital, Central South University
[3]University of Maryland, Baltimore County
[4]Brown University
[5]University of Pennsylvania
{ywang687,ydai55,craigj,hsair1,nloizou,whsu20,mimami1,hbai7}@jhu.edu
{pengjian666@csu,wujing622}@csu.edu
shenj@umbc.edu
zhichengjiao@brown.edu
paul.zhang2@pennmedicine.upenn.edu

## Abstract

Recent approaches to vision-language tasks are built on the remarkable capabilities of large vision-language models (VLMs). These models excel in zero-shot and few-shot learning, enabling them to learn new tasks without parameter updates. However, their primary challenge lies in their design, which primarily accommodates 2D input, thus limiting their effectiveness for medical images, particularly radiological images like MRI and CT, which are typically 3D. To bridge the gap between state-of-the-art 2D VLMs and 3D medical image data, we developed an innovative, one-pass, unsupervised representative slice selection method called Vote-MI, which selects representative 2D slices from 3D medical imaging. To evaluate the effectiveness of Vote-MI when implemented with VLMs, we introduce BrainMD, a robust, multimodal dataset comprising 2,453 annotated 3D MRI brain scans with corresponding textual radiology reports and electronic health records. Based on BrainMD, we further develop two benchmarks, BrainMD-select (including the most representative 2D slice of a 3D image) and BrainBench (including various vision-language downstream tasks). Extensive experiments on the BrainMD dataset and its two corresponding benchmarks demonstrate that our representative selection method significantly improves performance in zero-shot and few-shot learning tasks. On average, Vote-MI achieves a 14.6% and 16.6% absolute gain for zero-shot and few-shot learning, respectively, compared to randomly selecting examples. Our studies represent a significant step toward integrating AI in medical imaging to enhance patient care and facilitate medical research. We hope this work will serve as a foundation for data selection as vision-language models are increasingly applied to new tasks. Code and data examples are available at Github: https://github.com/YuliWanghust/BrainMD.

38th Conference on Neural Information Processing Systems (NeurIPS 2024) Track on Datasets and Benchmarks.

# 1 Introduction

Generalist foundation models, or large vision-language models (VLMs), such as GPT-4V [1], have revolutionized artificial intelligence by leveraging diverse large-scale datasets during pre-training. These models excel across multiple domains, including natural language processing and computer vision [33, 48, 55, 58, 18, 54], positioning them at the forefront of medical imaging advancements. However, a significant limitation of these state-of-the-art (SOTA) models is their restriction to 2D image input. This results in obstacles for their application to medical imaging, particularly with radiological images that are often 3D. To address these challenges, we propose a representative 2D slices selection approach called **Vote-MI**. This one-pass, unsupervised method selects representative 2D slices from 3D images, bridging the gap between SOTA VLMs and medical imaging. By employing Vote-MI, we aim to enhance diagnostic accuracy and automate medical reporting through the application of SOTA VLMs to 3D medical image analysis.

To thoroughly assess the effectiveness of our proposed Vote-MI method, a large, multimodal medical image dataset with paired textual data is essential. Therefore, we introduce **BrainMD**, a comprehensive dataset encompassing seven different types of brain tumors (details in Table 3). BrainMD includes 2,453 annotated 3D MRI brain scans, paired with textual data such as radiology reports, medical records, and demographic information. Based on BrainMD, we developed two benchmarks: **BrainMD-select and BrainBench**. BrainMD-select comprises the most representative 2D slices from the axial, sagittal, and coronal directions of 3D images, annotated by board-certified radiologists. BrainBench is derived from textual data and encompasses various tasks such as disease diagnosis and visual question answering. The dataset and its two benchmarks enable the evaluation of VLMs, ensuring robust model testing and accelerating advancements in AI-driven medical imaging diagnostics. Additionally, BrainMD and its associated benchmarks hold significant potential to benefit the broader research community by facilitating the future development of other 2D/3D VLMs.

Given that VLMs can perform downstream tasks with zero or few task demonstrations [31, 50], thereby eliminating the need for parameter updates, we evaluate the effectiveness of Vote-MI in VLMs through downstream task evaluations, including zero-shot and few-shot learning. Zero-shot testing [57] gauges the model's ability to tackle tasks without prior examples, utilizing its generalization capabilities from training data to novel tasks. Few-shot testing [51], or in-context learning, provides an alternative to traditional supervised tuning. In this study, we explore these capabilities in VLMs using the BrainMD dataset and Vote-MI method, thoroughly comparing model performance across random, Vote-MI, and radiologist-selected slices in zero-shot and few-shot scenarios.

Our experiments, over BrainMD and its two Benchmarks, demonstrate that our representative selective method substantially improves the VLM zero-shot and few-shot testing performance by balancing the diversity and representativeness of selected samples. For instance, Vote-MI achieves an average of 14.6% and 16.6% absolute gain for zero-shot and few-shot learning, respectively, compared to randomly selecting examples. Moreover, the improvement is consistent across different VLMs. Vote-MI representative selection also makes zero-shot learning and few-shot learning learning more stable and reduces the variance. Detailed results are shown in Section 6.

The code of Vote-MI and a few BrainMD examples are open-sourced. Our contributions are summarized as follows:

# 2 Related Works

## 2.1 Zero-shot and Few-shot Learning

Zero-shot learning (ZSL) [57] enables models to predict classes they haven't been explicitly trained on by leveraging auxiliary information. This approach addresses the challenge of acquiring labeled data for every class, especially in domains like rare medical diseases. Various ZSL methodologies include attribute-based [24], embedding-based [59], and generative approaches [30]. The versatility of ZSL has significant implications in vision and language processing, enhancing models' ability to generalize across diverse and unseen categories.

Few-shot learning [51] requires only a few annotated examples per test instance, avoiding the need for extensive fine-tuning. Recent research has proposed strategies to enhance few-shot learning, such as meta-training [34, 42], task instructions [26], and task formulation [22]. The selection of few-shot

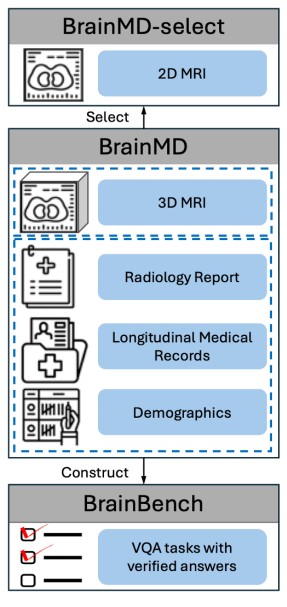

- **A large-Scale, multimodal longitudinal brain tumor dataset.** BrainMD (as shown in Figure 1) comprises 2453 studies from 561 patients, each including: (1) high-resolution 3D volumetric MRI images, (2) radiologist-selected representative slices in the coronal, sagittal, and axial directions, (3) paired radiology reports, and (4) structured longitudinal medical records. To our knowledge, this is the first dataset that combines 2D/3D medical images with both radiology reports and longitudinal medical records.

- **A novel representative slice selection method.** Vote-MI, a one-pass unsupervised representative slice selective method, is proposed to adapt SOTA 2D VLM for 3D medical image analysis. This method will enable the identification of the most representative 2D slices from complex 3D medical imaging datasets, allowing large VLMs to intake 3D imaging data by capturing the most important information from volumetric data.

- **Two Benchmarks for vision-language model evaluation.** Based on BrainMD, we establish two benchmarks for diagnosing and prognosticating outcomes of brain tumors. We conduct extensive evaluations, including zero-shot and few-shot testing scenarios, to assess VLM performance and capability.

Figure 1: Schematics of BrainMD Dataset and two benchmarks: BrainMD-select and BrainBench. The BrainMD dataset comprises MRI scans, paired with their radiology reports and medical records. BrainMD-select contains Radiologists selected representative slices. We curated diagnostic and prognostic labels based on the radiology reports and medical records to construct BrainBench.

examples is crucial, with studies questioning the necessity of correct labels. Our work introduces representative slice selection within a few-shot learning framework, emphasizing the importance of representative examples on the performance of VLMs.

## 2.2 Multi-modal Dataset

Publicly available medical datasets continue to drive significant advances in medical AI research [28, 4, 41, 14, 53]. However, very few currently available datasets are large-scale, multi-modal, and extensively labeled, particularly in medical domains that leverage both 2D and 3D medical images (Table 1). The limitations in data availability primarily stem from the inherent challenges associated with the release of medical data. Public sharing of medical data requires rigorous review processes to safeguard sensitive patient information from exposure [27]. Furthermore, the labeling process is often labor-intensive and costly [17, 61, 60].

Among previous contributions, the BRATS dataset [25] stands out as a large-scale work incorporating 3D MRI brain tumor images. However, BRATS lacks paired textual data and selected representative 2D slices. The BMs [38] and TCIA [52] datasets contain 3D MRI images with longitudinal medical record data. However, both have small dataset scales and do not include prognostic task labels or selected representative slices. Our BrainMD dataset addresses these gaps by introducing a large-scale dataset extracted from 2,453 brain tumor cases. It offers multiple modalities and labels, promising to enrich future research in this space.

## 2.3 Medical Multi-modal Vision Language Model

Recent research [33, 62] highlights the effectiveness of multimodal vision-language models (VLMs) in integrating image and text data for a variety of tasks. These models combine the perceptual capabilities of vision models [40, 47] with the generative power of large language models (LLMs) [43, 11, 13], gaining significant traction, particularly in medical image analysis. Existing medical VLMs [32, 49, 2, 37] often fine-tune publicly available 2D VLMs on medical image and text data to perform tasks such as image-text retrieval, report generation, and visual question answering. Models like LLaVA-Med [32], Med-PaLM-2 [49], and MedFlamingo [37] are derived from LLaVA [35], PaLM-E [12], and Flamingo [2], respectively.

Table 1: BrainMD vs. existing multimodal brain tumor medical image datasets

| Dataset | Modalities | | | | Counts | | Task Labels | |
|---|---|---|---|---|---|---|---|---|
| | Image | Slice | Report | EHR | Patients | Studies | Diagnosis | Prognosis |
| BRATS [25] | 3D MRI | × | × | × | 228 | many | × | × |
| Figshare [10] | 2D MRI | × | × | × | 3,064 | 3,064 | × | × |
| SARTAJ [7] | 2D MRI | × | × | × | 3,260 | 3,260 | × | × |
| BMs [38] | 3D MRI | × | × | ✓ | 75 | 637 | 2 | × |
| TCIA [52] | 3D MRI | × | × | ✓ | 47 | 156 | 2 | × |
| BrainMD | 3D MRI | ✓ | ✓ | ✓ | 561 | 2,453 | 2 | 1 |

However, these methods face challenges when applied to 3D medical images, such as CT and MRI scans, which contain rich spatial information. The common approach of slice-by-slice analysis is computationally expensive and often inadequate. While models like RadFM [55] support both 2D and 3D images, they primarily focus on text generation tasks like visual question answering (VQA) and generally underperform. More specialized VLMs, such as M3D-LaMed [5], Ct2rep [19], and Merlin [8], are designed specifically for 3D medical image analysis, tackling tasks like report generation and VQA, and pioneering vision question-answering tasks. Despite these advancements, 3D VLMs continue to struggle due to the lack of large, paired 3D image-report datasets and the high computational demands of model training.

## 3   Vote-MI: Representative slice selection method

In addressing the methodological challenges of transitioning from 2D to 3D medical images, particularly when employing SOTA VLMs, we propose the efficient, unsupervised Vote-MI method. This approach aims to efficiently identify highly diverse and representative 2D slices from 3D medical images in just one pass. Our method includes two main parts: 1) An unsupervised feature extraction process that operates directly on raw, unannotated images, and 2) A new criterion for assessing image diversity and representativeness during the selection process.

As shown in Figure 2, the representative selection pipeline consists of three major components: a) A patch-wise Variational Autoencoder (VAE) [39, 36] that serves as the unsupervised feature extraction network, effectively transposing each image sample into a low-dimensional feature descriptor; b) The Vote-MI algorithm, which identifies and selects a diverse and representative subset of images from the pool of unannotated data; and c) The VLMs, which are used downstream in various diagnostic and prognostic tasks. More details about the representative selection pipeline, including feature extraction network, Vote-MI representative slices selection, and other representative selection methods, are described in Appendix D.

## 4   Cohort Definition and Dataset Composition

Our study, approved by the Johns Hopkins University (Appendix A), identified 2,453 cases involving MRI brain tumor scans from 2010 to 2020. The cohort of MRI brain tumors was identified through a protocol involving random sampling, data cleaning, and inclusion criteria, resulting in a final cohort of 2,453 cases from 561 distinct patients (see Appendix C.1 for more details). Each of these images is de-identified. Summary statistics of the demographic characteristics of our final cohort are available in Table 2. Based on this cohort, we release the following as the BrainMD dataset:

- MRI images: The imaging slices for the BrainMD cohort in DICOM format.
- Radiology Report: The "Findings" and "Impression" section of the corresponding radiologist reports for all cases in the BrainMD cohort.
- Data From Medical Records: De-identified structured data from longitudinal records for each patient in our cohort, including diagnoses, procedures, lab results, and demographics.

A detailed description of the formatting and licensing details of BrainMD is in the Appendix C.2.

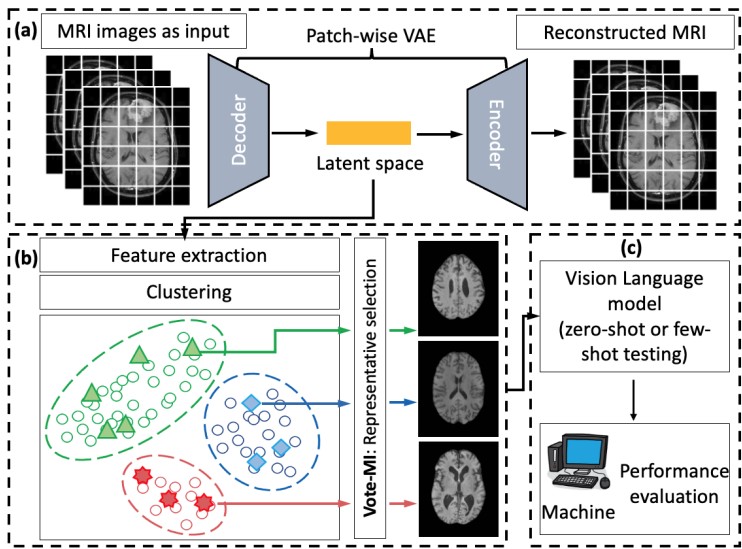

Figure 2: The workflow of our one-pass representative selection framework on brain MRI; (a) 2D patch-wise VAE for feature extraction; (b) Vote-MI: clustering-based representative selection; (C) downstream task evaluation using VLMs with zero-shot and few-shot scenarios.

Table 2: Demographic statistics of the proposed BrainMD dataset.

| Demographics Statistics | | |
|---|---|---|
| **Attributes** | | **All** |
| | Cases | 2,453 |
| | Patients | 561 |
| **Gender** | Female | 245 (43.6%) |
| | Male | 237 (42.2%) |
| | Unknown | 79 (14.2%) |
| **Age** | 0-18 | 38 (6.8%) |
| | 18-55 | 123 (21.9%) |
| | >55 | 387 (69.0%) |
| | Unknown | 13 (2.3%) |
| **Race** | White | 393 (70.0%) |
| | Asian | 24 (4.2%) |
| | Black | 29 (5.2%) |
| | Unknown | 115 (20.6%) |

# 5 Benchmark

To evaluate our proposed Vote-MI method and further demonstrate its effectiveness when applied to VLM downstream tasks, we developed two benchmarks: BrainMD-select and BrainBench. BrainMD-select, illustrated in Figure 1, is a 2D dataset created to evaluate the selection efficacy of Vote-MI. This dataset comprises the most representative 2D slices annotated by radiologists from the 3D BrainMD dataset. Furthermore, we introduce BrainBench, a benchmark designed to assess the performance of our representative slice selection method within SOTA VLMs, particularly under zero-shot and few-shot scenarios. BrainBench encompasses various tasks such as disease diagnosis, visual question answering, and even report generation.

These benchmarks, detailed in Appendix C.2, involve varied task formulations to allow comprehensive evaluations across different scenarios. Our experiments, as outlined in section 5.1 and 5.2, test the

efficacy of the Vote-MI selection method and its adaptability within VLMs under zero- or few-shot scenarios.

## 5.1 The effectiveness studies on the Vote-MI

To evaluate the Vote-MI method's effectiveness, we compare its output against four baselines: Uncertainty [23], Core-set [16], K-center [21], and Random sampling. Uncertainty selects instances where its confidence is lowest, indicating high uncertainty; Core set chooses data points furthest from the major data cluster, thus adding the most informative instances; K-center selects k data points as centers such that the maximum distance from any data point to its nearest center is minimized. We assess the performance of Vote-MI and these baselines using our BrainMD-select dataset, where radiologist selection is the gold standard. To ensure statistical validity, each method is run three times. An ablation study is also conducted to determine the contribution of each component in the Vote-MI method, which is shown in Appendix D.3.

## 5.2 Zero-shot and Few-shot Learning Tasks

We compare the performance of the VLMs, specifically Flamingo [2], Med-Flamingo [37], and Med-PaLM-2 [44] (due to our computation limitation) in a zero-shot setting using our custom benchmark, BrainBench (Figure 1). The models are evaluated on the following 2 diagnosis tasks:

- "Presence of cancer in the image (yes/no)? (**W/o cancer**)"
- "Name brain cancer types? (**Cancer types**)"

We adopt a bifurcated approach for few-shot learning (Figure 3). Initially, a representative subset is curated, selecting a finite collection of samples for labeling ahead of evaluation. Subsequently, for each test sample, pertinent examples are collated from this curated set, a step termed random prompt retrieval. The total labeling effort is demarcated by the number of samples curated and annotated in the preliminary phase. The second phase is constrained by the VLM's input capacity. Within these bounds, Vote-MI is recommended for its strategic selection of varied and indicative samples for selection. The model's performance is then assessed utilizing the BrainBench benchmark over two tasks as delineated in zero-shot learning and a new prognostic task as follows:

- "Describe the cancer status? (**Cancer status**)"

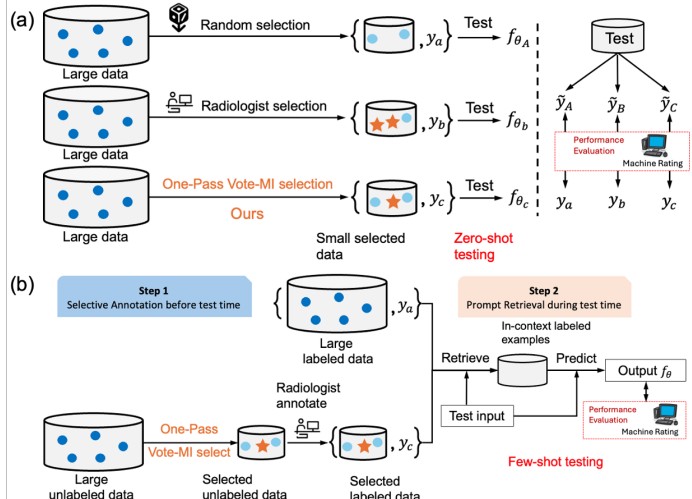

Figure 3: Schematic framework of downstream tasks for (a) zero-shot learning and (b) few-shot learning. In (a), we compare the one-pass Vote-MI representative 2D slice selection from 3D imaging with random and radiologist selections. For (b), the framework for few-shot learning utilizes Vote-MI, with alternatives being random or radiologist selections for the downstream task.

The statistical distribution of task labels for ground-truth of each sub-category ("W/o cancer", "Cancer types", and "Cancer status") is summarized in Table 3.

Table 3: Statistical analysis of three diagnostic and prognostic task labels in the BrainMD dataset, encompassing 2,453 cases and 561 patients. LGG: Low-Grade Gliomas; GBM: Glioblastoma Multiforme; CPTs: Choroid plexus tumors

| Task Labels Statistics | | | |
|---|---|---|---|
| **Types** | **Question** | **Answers** | **All** |
| | | | Cases 2,453 |
| | | | Patients 561 |
| **Diagnosis** | **W/o cancer** | Yes | 1714 (69.9%) |
| | | No | 381 (15.5%) |
| | | Uncertain | 357 (14.6%) |
| **Diagnosis** | **Cancer type** | GBM | 1326 (54.0%) |
| | | Glioma | 715 (29.1%) |
| | | LGG | 242 (9.8%) |
| | | Pineal tumors | 26 (1.1%) |
| | | Medulloblastoma | 72 (2.9%) |
| | | CPTs | 51 (2.0%) |
| | | Gangliocytoma | 20 (0.81%) |
| **Prognosis** | **Cancer status** | Improving | 38 (6.8%) |
| | | Progressing | 123 (21.9%) |
| | | No change | 387 (69.0%) |

## 5.3 Measuring Accuracy and Stability

**Accuracy:** These three tasks have predefined answer choices. Thus, we utilize accuracy (denoted as "ACC"), measuring the proportion of correctly identified cases to evaluate the VLM downstream tasks' performance.

**Stability:** Given a set of raw data, our Vote-MI representative slice selection method is not deterministic, with certain randomness. To assess the stability of Vote-MI and its impact on VLM performance, we conduct each experiment three times and average the results. Despite its non-deterministic nature, Vote-MI consistently enhances stability compared to other selection methods for both prognostic and diagnostic tasks.

# 6 Results and Analysis

## 6.1 Effectiveness studies of Vote-MI

Table 4 summarizes the slice selection accuracy of different methods on the BrainMD dataset and BrainMD-select benchmark. The accuracies are calculated with a slice number error tolerance of $\pm$ 5, given the nature of brain tumor images where multiple representative slices can capture the characteristics of the tumors. The Vote-MI method achieved the highest accuracy at 59.4% and the lowest variance $\pm$ 4.2%, which is statistically significantly better than the other baseline methods.

Table 4: Performance comparison of different selection methods on BrainMD dataset.

| | **Uncertainty** | **Core-set** | **K-center** | **Random** | **Vote-MI** |
|---|---|---|---|---|---|
| **Accuracy** | 52.2 ($\pm$ 4.9) | 53.5 ($\pm$ 7.0) | 47.6 ($\pm$ 5.4) | 28.2 ($\pm$ 8.6) | 59.4 ($\pm$ 4.2) |

## 6.2 Zero-shot Learning Results

As shown in Figure 4 are our results from zero-shot learning over the BrainMD dataset with two downstream tasks including: 1) with or without cancer (binary classification) and 2) identifying cancer type (vision question answering). Over all datasets, the Vote-MI representative selective method outperforms the random baseline by a large margin (23.5% absolute gain on average in w/o cancer and 14.4% absolute gain on average in cancer type) under the zero-shot scenario.

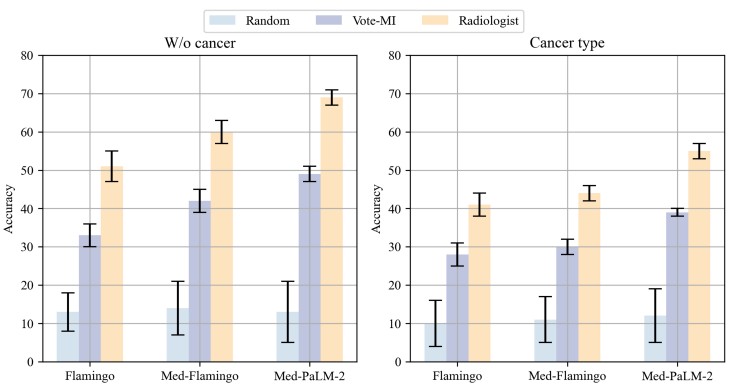

Figure 4: Performance comparisons of various VLMs under two disease diagnostics downstream task settings. Compared to random selection, the Vote-MI representative selection method consistently improves the performance with a large margin of zero-shot learning with pre-trained VLMs.

## 6.3 Few-shot Learning Results

Table 5: Few-shot learning results with random, radiologist, and Vote-MI selection methods on the BrainMD dataset, with a selection budget of 2, 4, or 8. Across the board, representative selection with Vote-MI substantially outperforms the random selection baseline for few-shot learning. Further, Vote-MI largely reduces the variance over three trials (see the results in Appendix D.3), making few-shot learning more stable. $\Delta$ means absolute gain between Random and Vote-MI.

| Size | Method | Tasks | | |
|---|---|---|---|---|
| $\mathcal{|L|}$ | Selection | W/o cancer | Cancer types | Cancer status |
| 2 | Random | 21.3 ($\pm$ 5.6) | 18.3 ($\pm$ 7.1) | 19.6 ($\pm$ 4.0) |
| 2 | Vote-MI | 43.5 ($\pm$ 2.3) | 35.3 ($\pm$ 4.5) | 37.7 ($\pm$ 3.8) |
| 2 | Radiologist | 64.5 ($\pm$ 3.1) | 55.3 ($\pm$ 4.2) | 48.4 ($\pm$ 3.9) |
| 2 | $\Delta$ Absolute gain | +22.2 | +17.0 | +18.1 |
| 4 | Random | 28.1 ($\pm$ 5.1) | 24.5 ($\pm$ 6.2) | 26.7 ($\pm$ 5.8) |
| 4 | Vote-MI | 50.7 ($\pm$ 2.8) | 43.0 ($\pm$ 3.9) | 42.7 ($\pm$ 4.0) |
| 4 | Radiologist | 68.2 ($\pm$ 3.4) | 60.2 ($\pm$ 4.5) | 52.7 ($\pm$ 4.3) |
| 4 | $\Delta$ Absolute gain | +22.6 | +19.5 | +16.0 |
| 8 | Random | 30.1 ($\pm$ 6.1) | 28.7 ($\pm$ 6.5) | 30.4 ($\pm$ 6.3) |
| 8 | Vote-MI | 55.3 ($\pm$ 3.1) | 46.0 ($\pm$ 4.2) | 46.7 ($\pm$ 4.1) |
| 8 | Radiologist | 70.1 ($\pm$ 3.5) | 62.3 ($\pm$ 4.3) | 56.9 ($\pm$ 4.4) |
| 8 | $\Delta$ Absolute gain | +15.2 | +18.3 | +16.3 |

In this study, we perform an extensive analysis of few-shot learning to provide further guidance, examining representative slice selection from multiple dimensions: varying VLMs, different downstream tasks, and selection sizes. Our findings from the BrainMD dataset, detailed in Table 5, show results for selection budgets ranging from 2, 4 to 8. This range accommodates the input limits of VLMs, allowing full integration of examples into prompts without additional sampling. Across all tasks and VLMs, the Vote-MI method for selecting representative slices significantly outperforms a random baseline for all selection sizes, with a standout 16.6% average absolute gain when the set

size is 8 (see Figure 5). Notably, using just two Vote-MI selected examples achieves better outcomes than eight randomly chosen ones across all tasks, highlighting the effectiveness of strategic example selection in few-shot learning.

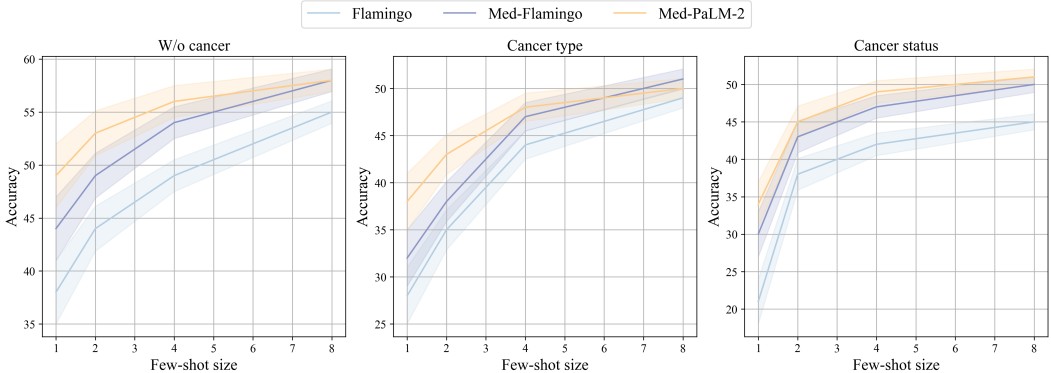

Figure 5: Few-shot testing comparison among three different downstream tasks. Vote-MI representative slice selection improves the performance of zero-shot learning with pre-trained VLMs.

## 6.4 Impact of Vote-MI on Stability

As shown above, our extensive experiments demonstrated that representative selection yields higher accuracy across all downstream tasks and is crucial for the VLMs' success in both zero-shot and few-shot learning. However, Vote-MI is not a deterministic representative selection method and includes certain randomness, conditioned on a set of unlabeled image samples and a selection budget. Thus, we explored the stability of the method. From our experiments, we observed a reduction in variance for both zero-shot and few-shot learning across all downstream tasks. Therefore, the variance of Vote-MI arises solely from how the unlabeled samples are collected, significantly improving the robustness of zero-shot and few-shot learning. We, therefore, recommend that researchers and practitioners use representative slice selection methods (e.g., our Vote-MI method) to better benefit from the zero-shot and few-shot learning capabilities of VLMs with increased stability.

## 7 Limitation and Conclusion

### 7.1 Limitations

First, BrainMD contains data from only a single site, and the Vote-MI representative selection model or other future models trained on BrainMD may not generalize to other patient populations. Second, although labels are assigned based on large language model output and manually reviewed by radiologists, there still might be inaccuracies in some cases. Finally, although the effectiveness of Vote-MI improved significantly compared to random and other slice selection methods, the downstream task performance is still statistically significantly worse than the radiologist's selection. We hope our paper can serve as a baseline and inspire further research on methods for representative selection, bridging the gap between 2D vision language models and 3D medical image data.

Future efforts will focus on optimizing the representative selection framework to further improve accuracy. This includes researching potentially better feature extraction networks. Given the relatively high homogeneity within tumor lesions, generative-based (e.g., diffusion probabilistic models [45, 56]) or contrastive-based unsupervised learning methods [9, 20] may be more effective and accurate in doing the feature extraction. Additionally, new criteria for assessing image diversity and representativeness are needed. Beyond the mutual information metrics used in our paper, graph-based metrics [46, 6] and confidence-based scoring [29] could potentially enhance selection accuracy and, consequently, the VLM's downstream task performance.

## 7.2 Conclusion

There are three main contributions to this work. First, we present BrainMD, a large-scale medical dataset with multiple modalities, comprising health records of 2,453 high-quality MRIs from 561 patients, complete with radiology reports, and structured data from medical records. Second, we propose a novel one-pass unsupervised representative slice selection method, Vote-MI, to select representative 2D slices from 3D volumetric data, bridging the gap between current vision-language models and their application to medical images. Third, we use this dataset to create two benchmarks, BrainMD-select and BrainBench. Using these benchmarks, we conducted in-depth studies on our proposed Vote-MI method. In terms of task performance, Vote-MI significantly improves performance across three diverse tasks. In conclusion, this work has laid the foundation for future research into representative slice selection methods for analyzing 3D medical imaging data with VLMs that only take 2D input. By openly sharing BrainMD, we hope to spark new advances in this critical area of healthcare.

## Acknowledgments and Disclosure of Funding

This publication was made possible by the Johns Hopkins Institute for Clinical and Translational Research (ICTR), which is funded in part by Grant Number 1UM1TR004926-01 from the National Center for Advancing Translational Sciences (NCATS) a component of the National Institutes of Health (NIH), and NIH Roadmap for Medical Research. Its contents are solely the responsibility of the authors and do not necessarily represent the official view of the Johns Hopkins ICTR, NCATS or NIH.

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
