# A    Data User Agreement

The development of BrainMD and its associated two benchmarks are approved by the Institutional Review Board IRB of the Johns Hopkins University (JHU). All the data were obtained through the study "Enhancing large vision-language model efficacy through representative slice selection from 3D medical imaging data" (IRB Number: 00452757). The full dataset is under restricted access, and interested parties should contact the Principal Investigator (PI) to obtain the DUA required for full access.

An example of detailed terms is as follows; you may also request the complete DUA from the PI:

- Provider shall provide the data set (the "Data") to Recipient for the research purpose (the "Project"). Provider shall retain ownership of any rights it may have in the Data, and Recipient does not obtain any rights in the Data other than as set forth herein.

- Recipient shall not use the Data except as authorized under this Agreement. The Data will be used solely to conduct the Project and solely by Recipient Scientist and Recipient's faculty, employees, fellows, students, and agents ("Recipient Personnel") and Collaborator Personnel that have a need to use, or provide a service in respect of, the Data in connection with the Project and whose obligations of use are consistent with the terms of this Agreement (collectively, "Authorized Persons").

- Except as authorized under this Agreement or otherwise required by law, Recipient agrees to retain control over the Data and shall not disclose, release, sell, rent, lease, loan, or otherwise grant access to the Data to any third party, except Authorized Persons, without the prior written consent of Provider. Recipient agrees to establish appropriate administrative, technical, and physical safeguards to prevent unauthorized use of or access to the Data and comply with any other special requirements relating to safeguarding of the Data.

- Recipient is encouraged to make publicly available the results of the Project. Before Recipient submits a paper or abstract for publication or otherwise intends to publicly disclose information about the results of the Project, the Provider will have thirty (30) days from receipt to review proposed manuscripts and ten (10) days from receipt to review proposed abstracts to ensure that the Data is appropriately protected. Provider may request in writing that the proposed publication or other disclosure be delayed for up to thirty (30) additional days as necessary to protect proprietary information.

# B    Data De-identification

All BrainMD data, including high-resolution 3D MRI scans, radiologist-selected 2D MRI slices, paired radiology reports, and longitudinal medical records, are manually reviewed to ensure that any protected health information (PHI) is removed before being applied to the vision-language model for downstream tasks.

Defacing is an important step in the preprocessing of brain MRI data to ensure the ethical handling, sharing, and publication of such data while maintaining high standards of privacy. The MRI defacing method, as described in `https://doi.org/10.1016/j.neuroimage.2021.117845`, is applied to each image before the data is shared and before it is utilized for downstream tasks in the VLMs. We provide a preview subset of the entire dataset for reviewers as shown in Github: `https://github.com/YuliWanghust/BrainMD`.

As the authors of the submitted dataset and corresponding manuscript, we affirm that we take full responsibility for its content. We ensure that this dataset and manuscript are original and that all data collection procedures were carried out ethically, respecting all relevant rights and regulations. We confirm that we have obtained all necessary permissions for the use of the data included in the dataset and that the data does not infringe upon any existing copyright, proprietary, or personal rights of others.

## C    Chohort definition and Dataset documents

### C.1    Cohort Definition

The flowchart of our cohort definition protocol is illustrated in Figure 6. In the final phase of cohort construction and data cleansing, we excluded cases lacking a report or an impression section. This step narrowed the dataset down to 3,708 cases eligible for further scrutiny. We proceeded to select appropriate MRI series for each study, focusing on those with T1-weighted and T2 FLAIR-weighted MRI sequences and a slice thickness ranging from 1.0 mm to 5.0 mm. MRIs comprising fewer than 50 slices were also discarded. This selection process yielded 2,579 studies. Subsequently, we removed duplicate reports or MRI images and streamlined the final selection phase by eliminating redundant cases. The ultimate dataset comprises 2,453 brain tumor studies from a total of 561 unique patients.



Figure 6: Framework of our final cohort definition process.

### C.2    Dataset documents: BrainMD format

We detail and define our cohort in the data using a master cohort CSV file (brainMD.csv), which includes the following primary columns:

- Patient ID: The de-identified patient ID
- Time Stamps: The date of the MRI procedure

Each case in the dataset is characterized by a textual component that includes a Patient ID/Time Stamps pair, along with demographic data such as age, gender, and race. Additionally, there are multiple columns designated for vision language task labels. Each label undergoes manual verification by a board-certified neurologist.

The imaging component of each case, comprising MRI scans from BrainMD, is delivered in NifTI format accompanied by a JSON file. To maintain patient confidentiality, the patient ID and study date are anonymized. The dataset includes only selected DICOM header information from the original DICOM files: ["MagneticFieldStrength", "ImagingFrequency", "Manufacturer", "MRAcquisition-Type", "ScanningSequence", "SeriesNumber", "SliceThickness": 3, "SpacingBetweenSlices", "SAR", "EchoTime", "RepetitionTime"]. Further statistical details are available in the Appendix. C.2.1.

#### C.2.1    BrainMD MRI statistics

Based on our inclusion criteria in Appendex C.1, each MRI study can have slices over 50. The BrainMD studies range from 1.00 mm to 5.00 mm (Table 6) collected from MRI scanners by 5 different manufacturers (Table 7).

Table 6: Statistics of BrainMD scans slices thickness distribution

| Slice Thickness (mm) | Count |
|---|---|
| 1.00 | 757 |
| 2.00 | 94 |
| 3.00 | 864 |
| 4.00 | 158 |
| 5.00 | 580 |

Table 7: Statistics of BrainMD scans manufacturers distribution

| Manufacturer | Count |
|---|---|
| Siemens | 2281 |
| GE Medical Systems | 144 |
| Philips | 22 |
| Toshiba | 4 |
| Hitachi | 2 |

### C.3 Dataset documents: BrainMD-select format

Similar to the BrainMD format, the BrainMD-select will be a master cohort CSV file (BrainMD-select.csv), which includes the following primary columns. Both textual labels and 2D NifTI format with an associated JSON file are contained in BrainMD-select. All 2D images are representative slices that were manually selected by board-certified neurologists to ensure they best represent the brain tumor from the original 3D volumetric data.

### C.4 Dataset documents: BrainBench format

First, we include two brain tumor diagnostic label columns. "W/o cancer" is the label used in all of our experiments for diagnosing the presence or absence of a brain tumor, described in detail in Appendix C.5. Every case in our cohort is assigned either "True" or "False" for this label. "Cancer type" is the other label column that further specifies the types of cancer. There are seven types of brain cancer included in this label: Glioblastoma Multiforme (GBM), Glioma, Low-Grade Gliomas (LGG), Pineal tumors, Medulloblastoma, Choroid Plexus Tumors (CPT), and Gangliocytoma.

Second, we include one prognostic label column. These label columns correspond to the status of the brain tumor for each case during longitudinal MRI scans. Every case in our cohort is assigned either "Improving," "Progressing," or "No change" for this column.

### C.5 Dataset documents: Tasks label definition

As part of our project, we developed and validated a set of diagnostic labels for brain tumors and a prognostic label for monitoring the progress of brain tumors in each case within our dataset. Board-certified neurologists manually annotated the ground truth for these labels, utilizing MRI scans and radiology reports as references.

- **W/o cancer:** This label is crucial as it identifies the absence of brain tumors, which are potentially life-threatening conditions characterized by abnormal and uncontrolled cell growth within the brain or its surrounding structures. Early and accurate diagnosis can significantly impact treatment options and patient outcomes.

- **Cancer types:** This label indicates the specific types of brain tumors, which is essential for tailoring patients' treatment, as different cancer types require different treatment protocols. Precise classification of brain tumor types is vital for precision medicine.

- **Cancer status:** Our BrainMD dataset is a longitudinal dataset where patients undergo multiple scans over an extended period, covering both pre-surgery and post-surgery phases, to monitor the progress of brain tumors. We leveraged this characteristic to develop a prognostic task. This prognostic task benefits clinicians by tracking tumor progression, evaluating the effectiveness of treatments, and making informed decisions about future medical interventions, ultimately improving patient outcomes and quality of care.

Using these image scans, reports, and task labels, we developed and tested a representative slice selection method to bridge the gap between 3D volumetric images and current state-of-the-art 2D vision-language models (VLMs). This dataset is the first of its kind to provide both 2D and 3D brain tumor images along with corresponding reports. We believe it will benefit not only the representative slice selection method and 2D vision-language models but also the development of 3D VLMs. The statistical analysis of the task labels is summarized as follows.

### C.5.1 Task label distribution

We constructed three sets of task labels, including two diagnostic tasks and one prognostic task ("W/o cancer", "Cancer types", and "Cancer status"). The statistical distribution of task labels for ground-truth of each sub-category is summarized in Table 8.

Table 8: Statistical analysis of three diagnostic and prognostic task labels in the BrainMD dataset, encompassing 2,453 cases and 561 patients. LGG: Low-Grade Gliomas; GBM: Glioblastoma Multiforme; CPTs: Choroid plexus tumors

| Task Labels Statistics | | | |
|---|---|---|---|
| **Types** | **Question** | **Answers** | **All** |
| | | | Cases 2,453 |
| | | | Patients 561 |
| **Diagnosis** | **W/o cancer** | Yes | 1714 (69.9%) |
| | | No | 381 (15.5%) |
| | | Uncertain | 357 (14.6%) |
| **Diagnosis** | **Cancer type** | GBM | 1326 (54.0%) |
| | | Glioma | 715 (29.1%) |
| | | LGG | 242 (9.8%) |
| | | Pineal tumors | 26 (1.1%) |
| | | Medulloblastoma | 72 (2.9%) |
| | | CPTs | 51 (2.0%) |
| | | Gangliocytoma | 20 (0.81%) |
| **Prognosis** | **Cancer status** | Improving | 38 (6.8%) |
| | | Progressing | 123 (21.9%) |
| | | No change | 387 (69.0%) |

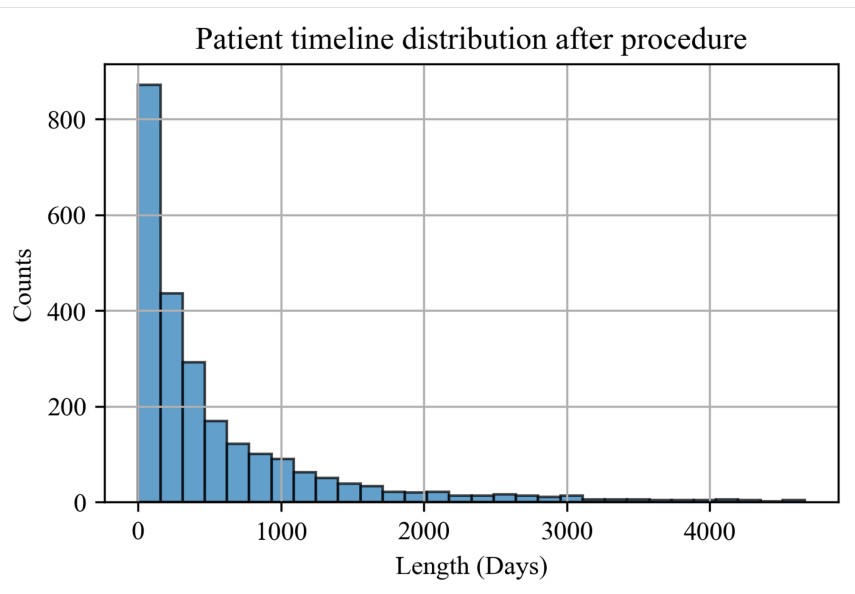

Figure 7: Patient timeline distribution for all cases in BrainMD dataset.

**Distribution of follow up-times.** Based on the longitudinal medical records in our dataset, we have a substantial history of follow-up times for each patient. Figure 7 summarizes the distribution of follow-up times in days. The follow-up times in the entire dataset range from 0 to 6,887 days, with an average of 448.19 days and a standard deviation of 764.87 days, as summarized in Table 9.

Table 9: Statistics of BrainMD scans intervals across all patients and studies

| | Min | Max | Average | Standard deviation |
|---|---|---|---|---|
| Interval (in days) | 0 | 4667 | 555.06 | 756.98 |

# D  Vote-MI extra results

## D.1  Vote-MI algorithm details

The pseudo algorithm of the Vote-MI is shown in Algorithm 1, containing the feature extraction, sampling, intra-cluster selection, and inter-cluster selection. The code implementation details including MCMC, DBSCAN, and Vote-MI selection are summarized as follows.

### D.1.1  Feature Extraction Network

The first step in our approach is to extract features from input raw images in an unsupervised manner. To this end, we employ a patch-wise VAE as the feature extraction method, mapping the input data to a low-dimensional latent feature space. The VAE consists an encoder that maps an input image sample $x$ to a latent space representation $z$, and a decoder that reconstructs the sample in the original space from the latent representation. The VAE minimizes the difference between a posterior distribution and the distribution of latent neurons, with the difference measured by the Kullback-Leibler divergence.

### D.1.2  Vote-MI: Representative Slices Selection

The goal of the Vote-MI stage is to select a diverse and representative 2D slice set, $X_r$, from the entire input 3D image set, $X_u$, as suggested samples for VLM downstream task input. To achieve this, we propose a two-stage method that combines clustering-based and mutual information-based selection methods, which is shown in Appendix Algorithm 1.

In the first stage, we conduct DBSCAN [15] clustering on the low-dimensional VAE latent space to determine the proper number of clusters, $K$. Then, we utilize MCMC [3] as the sampling technique to create an initial subset for each cluster while adhering to a predetermined budget. This selected subset serves as the initial candidate set, upon which the subsequent intra-cluster selection process based on mutual information builds. Using the initial subset as a starting point, we then apply the greedy strategy to select a set of images $X_k$ from each cluster $C_k$. In the second stage, we apply mutual information-based selection again. We select the most representative image from all the $X_k$'s one at a time to form the final $X_r$. As a result, intra-cluster redundancy is reduced in the first stage, and inter-cluster redundancy is reduced in the second stage while maintaining inter-cluster diversity.

Specifically, we use mutual information $\texttt{MI}(x_i, x_j)$ between two images $x_i$ and $x_j$ as a metric to measure how well one represents the other. A larger value represents better representativeness of one to the other. We extend the concept of representativeness from image-image to image-set: given an image $x$ and a set of images $X$, we have

$$\texttt{MI}(x, X) = \max_{x_i \in X} \texttt{MI}(x, x_i). \tag{1}$$

Similarly, we extend the definition of representativeness from image-set to set-set, denoted as $F$:

$$F(X, C) = \sum_{x_i \in X} \texttt{MI}(x_i, C), \tag{2}$$

which we also refer to as coverage score, since it shows how well set $X$ represents set $C$.

In the first stage, for each cluster $C_k$, we choose a sample $x$ each iteration that best "represents" $C_k$:

$$x = \operatorname*{argmax}_{x \in C_k}(F(X_k \cup x, C_k) - F(X_k, C_k)), \tag{3}$$

remove $x$ from $C_k$ and add it to $X_k$ till $X_k$ reaching the predefined size $\delta$.

In the second stage, the candidate set is defined as $X_c := \cup_{k=1}^{K} X_k$, from which we form the final set $X_r$. We use $\theta$ to control the sizes of $X_r$. Iteratively, we choose one image sample $x^*$ from $X_c$ and

**Algorithm 1** Vote-MI: Patch-wise representative selection algorithm

---

**Require:** Unannotated image patch set $X_u$, intra-cluster size control $\delta$, inter-cluster size control $\theta$, final diverse and representative image-set $X_f = \emptyset$, final diverse and representative image patch-set $X_r = \emptyset$

**Ensure:** Selected small image-set for downstream $X_f$

 1: Train VAE on $X_u$ to get feature vectors
 2: Apply DBSCAN on feature vectors to get clusters $C_i$ for $i = 1, ..., K$
 3: **for** each cluster $C_i$ **do**
 4:     Sample 20% images from $C_i$ to initialize $X_i$          *$^*$MCMC sampling*
 5:     $C_i = C_i \setminus X_i$ to exclude $X_i$ from $C_i$
 6:     **while** $|X_i|/|C_i| < \delta$ **do**      *$^*$intra-cluster selection*
 7:         Calculate and rank the coverage score $F$, by selecting $x$ from $C_i$ that maximizes $F(X_i \cup x, C_i) - F(X_i, C_i)$, as shown in Eq.3
 8:         Add $x$ to $X_i$ and remove $x$ from $C_i$
 9:     **end while**
10:     Add every $X_i$ together to form the candidate set $X_c$
11: **end for**
12: **while** $|X_r|/|X_u| < \theta$ **do**        *$^*$inter-cluster selection*
13:     Calculate and rank the coverage score $F$, by selecting $x$ from $X_c$ that minimizes $F(X_r \cup x, X_c) - F(X_r, X_c)$, as shown in Eq.4
14:     Add $x$ to $X_r$ and remove $x$ from $X_c$
15: **end while**
16: **for** EACH $x_r$ in $X_r$ **do**        *$^*$ranking the representative slice using the selected patches*
17:     $SliceNo. = GetSliceNo.(x_r)$
18:     $direction = get_d direction(x_r)$
19:     **while** $direction = Axial$ or $Coronal$ or $Sagittal$ **do**
20:         **while** $CountVotes(X_f[SliceNo.]) < CountVotes(x_r)$ **do**
21:             $X_f[SliceNo.] = x_r$
22:         **end while**
23:     **end while**
24: **end for**
        **return** $X_f$

---

put it in $X_r$, that has the lowest coverage score:

$$x^* = \underset{x \in X_c}{\mathrm{argmin}}(F(X_r \cup x, X_c) - F(X_r, X_c)) \tag{4}$$

remove $x^*$ from $X_c$ and add it to $X_r$ till $X_r$ reaching the predefined size $\theta$.

The selection of the images essentially sorts the image samples in $X_c$ based on their diversity and representativeness. Finally, we can get representative image samples according to the order, until a pre-defined selection budget ratio is reached (e.g. 10%). This two-stage process ensures that the selected samples are both diverse and representative, leading to efficient representative slice selection.

### D.1.3 Random and other Representative Selection Methods

To quantify the effect of Vote-MI, we also provide random and other baselines. Once we have a set of representative examples from different representative selection methods, we randomly retrieve a few examples with a setting budget (e.g. 1, 2, 4, and 8) from the representative set as zero-shot and few-shot learning examples for each instance. In section 6, we show that this baseline substantially underperforms the Vote-MI method, demonstrating the importance of the representative selection step to improve the VLM's performance on medical image analysis. Results comparing other representative selection methods can be found in Appendix D.2.

### D.2 Effectiveness of Vote-MI

Here is a summary of the results demonstrating the effectiveness of Vote-MI compared to random selection methods. Figures 8 and 9 illustrate the locations of selected representative slices and the distribution of selected slices among Random, Vote-MI, and radiologist selections across the entire

BrainMD dataset. Compared to the random selection method, we observed a closer match in slice numbers and slice distribution between Vote-MI and radiologist selections. Although the selection results are not perfect, Vote-MI shows significantly lower variance and error compared to random selection.

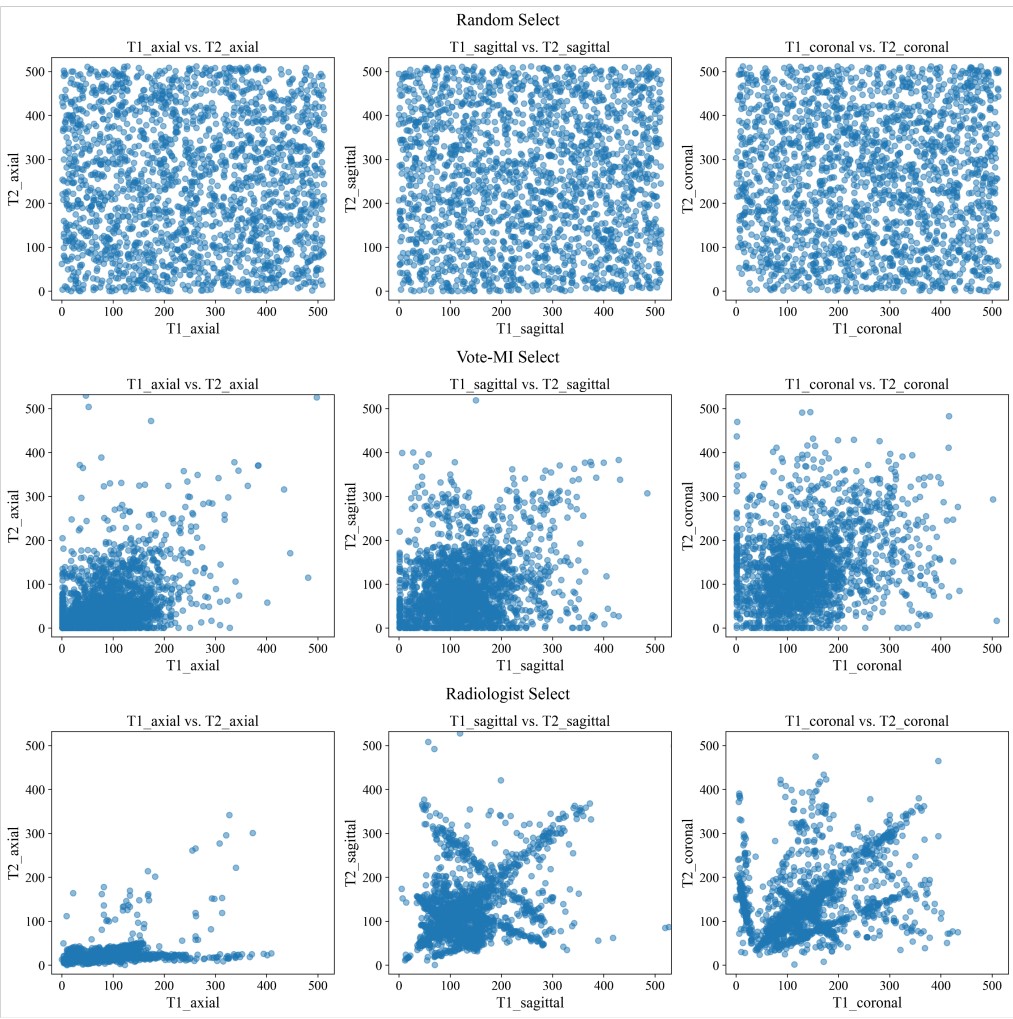

Figure 8: Summary of the distribution of selected representative slices for the BrainMD dataset, categorized by selection method: Random (top), Vote-MI (middle), and Radiologist (bottom). The T1 slice locations are plotted on the X-axis and T2 slice locations on the Y-axis.

We have already reported the slice selection accuracy of different methods on the BrainMD dataset using the BrainMD-select as the benchmark in section 6.1. The results as follows summarize the downstream task performance using the selected representative slices from different representative selection methods under zero-shot and few-shot learning. As shown in Figure 10, the VLMs' zero-shot learning accuracy performance comparison is summarized among different representative selection methods, including Random, K-center, Uncertainty, Core-set, and Vote-MI. Vote-MI representative selection method consistently improves performance by a large margin in zero-shot learning with pre-trained VLMs.

Here, we analyze representative selection for few-shot learning. See in Table 10 is our few-shot learning results from the BrainMD dataset with selection budgets of $|\mathcal{L}| \in (2, 4, 8)$. 8 is chosen due to the input limitation of VLMs so that all annotated examples can be directly fit to the prompt for the vision language models without further sampling. Over all downstream tasks and VLMs, the

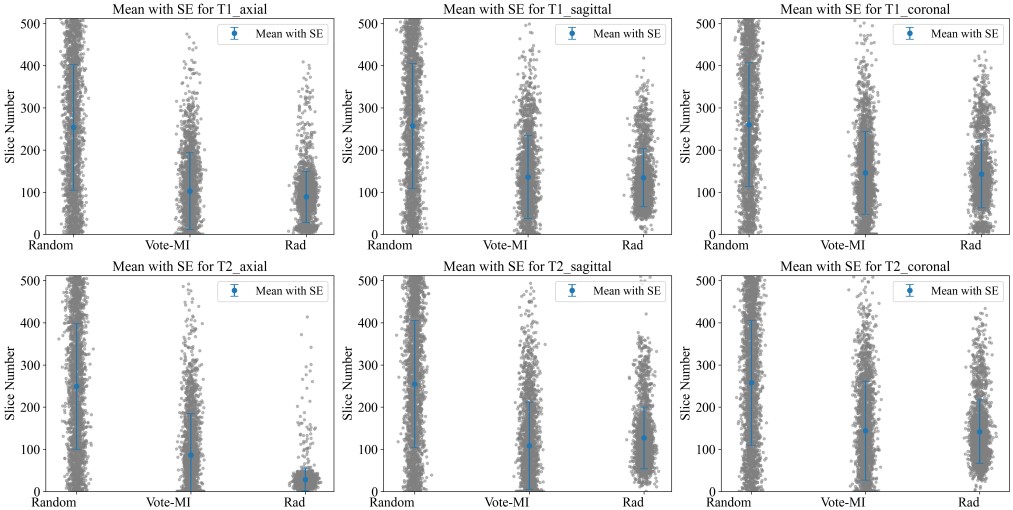

Figure 9: Location distribution summary of the selected representative slices among Random, Vote-MI, and radiologist across the entire BrainMD dataset. T1 and T2 images for axial, sagittal, and coronal directions are summarized in the figures. SE stands for standard errors.

Vote-MI representative slice selection method outperforms all other slice selection methods (including Random, K-center, Uncertainty, and Core-set) for all selection budgets.

Table 10: In-context learning results with Random-selected, K-center, Uncertainty, Core-set, and Vote-MI representative selection methods on the BrainMD dataset, with a selection budget of 2, 4, or 8. Across the board, representative selection with Vote-MI substantially outperforms the randomly selected baseline for few-shot learning.

| Size | Method | Tasks | | |
|------|--------|-------|---|---|
| $\|\mathcal{L}\|$ | Selection | W/o cancer | Cancer types | Cancer status |
| 2 | Random | 21.3 ($\pm$ 5.6) | 18.3 ($\pm$ 7.1) | 19.6 ($\pm$ 4.0) |
| 2 | K-Center | 34.1 ($\pm$ 3.2) | 28.9 ($\pm$ 5.0) | 31.0 ($\pm$ 3.9) |
| 2 | Uncertainty | 32.8 ($\pm$ 3.5) | 27.5 ($\pm$ 4.8) | 29.4 ($\pm$ 3.7) |
| 2 | Core-set | 36.4 ($\pm$ 3.1) | 30.7 ($\pm$ 4.7) | 32.8 ($\pm$ 3.5) |
| 2 | Vote-MI | **43.5 ($\pm$ 2.3)** | **35.3 ($\pm$ 2.5)** | **37.7 ($\pm$ 2.8)** |
| 4 | Random | 28.1 ($\pm$ 5.1) | 24.5 ($\pm$ 6.2) | 26.7 ($\pm$ 5.8) |
| 4 | K-Center | 39.6 ($\pm$ 3.7) | 32.9 ($\pm$ 4.3) | 35.3 ($\pm$ 4.2) |
| 4 | Uncertainty | 40.7 ($\pm$ 3.4) | 33.8 ($\pm$ 4.1) | 36.0 ($\pm$ 3.9) |
| 4 | Core-set | 41.8 ($\pm$ 3.6) | 34.6 ($\pm$ 4.0) | 37.2 ($\pm$ 3.8) |
| 4 | Vote-MI | **50.7 ($\pm$ 2.8)** | **43.0 ($\pm$ 2.9)** | **42.7 ($\pm$ 2.0)** |
| 8 | Random | 30.1 ($\pm$ 6.1) | 28.7 ($\pm$ 6.5) | 30.4 ($\pm$ 6.3) |
| 8 | K-Center | 48.6 ($\pm$ 3.4) | 39.9 ($\pm$ 4.5) | 41.5 ($\pm$ 4.3) |
| 8 | Uncertainty | 49.2 ($\pm$ 3.5) | 40.4 ($\pm$ 4.4) | 42.0 ($\pm$ 4.2) |
| 8 | Core-set | 50.1 ($\pm$ 3.3) | 41.2 ($\pm$ 4.3) | 43.1 ($\pm$ 4.1) |
| 8 | Vote-MI | **55.3 ($\pm$ 3.1)** | **46.0 ($\pm$ 2.2)** | **46.7 ($\pm$ 2.1)** |

### D.3 Ablation studies on Vote-MI

An ablation study is conducted to determine the contribution of each component in the Vote-MI method, with random sampling as the baseline, we will compare output accuracy under different conditions:

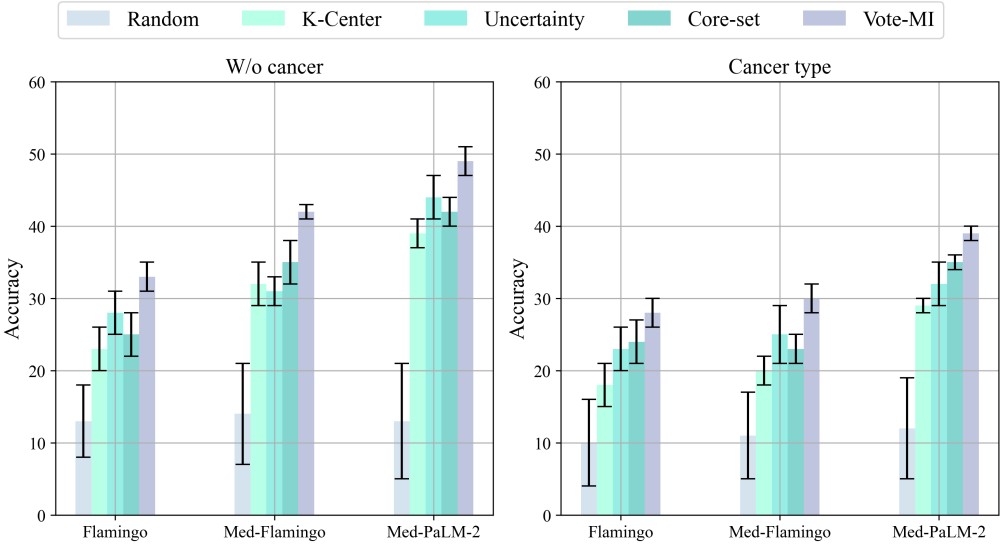

Figure 10: Performance comparisons of various VLMs under two disease diagnostics downstream task settings using different slice selection methods, including Random, K-center, Uncertainty, Core-set, and Vote-MI, show that the Vote-MI representative selection method consistently improves performance by a large margin in zero-shot learning with pre-trained vision-language models.

- Initialize with Random sampling
- Replace Random sampling with MCMC sampling
- Implement DBSCAN clustering before MCMC sampling
- Integrate the Vote-MI selective method

The outcomes of the ablation studies will be used to ascertain the individual contribution and effectiveness of each component within the Vote-MI framework. The detailed results of the ablation study are summarized in Figure 11 for zero-shot learning and Table 11 for few-shot learning.

For the zero-shot testing scenario, the baseline was set as random sampling. When we replaced random sampling with MCMC sampling while keeping the same VLMs, we observed a 1.4% improvement in accuracy. The addition of DBSCAN clustering before MCMC sampling led to a further 4.2% improvement. Incorporating the Vote-MI selective annotation method yielded an additional 12.1% improvement in accuracy. These findings highlight the effectiveness of each component of the Vote-MI method and its overall superiority over the baseline methods.

For the few-shot testing scenario, the baseline was set as random sampling. When we replaced random sampling with MCMC sampling while keeping the same VLMs, we observed a 1.9% improvement in accuracy. The addition of DBSCAN clustering before MCMC sampling led to a further 5.1% improvement. Incorporating the Vote-MI selective annotation method yielded an additional 10.1% improvement in accuracy. These findings highlight the effectiveness of each component of the Vote-MI method and its overall superiority over the baseline methods.

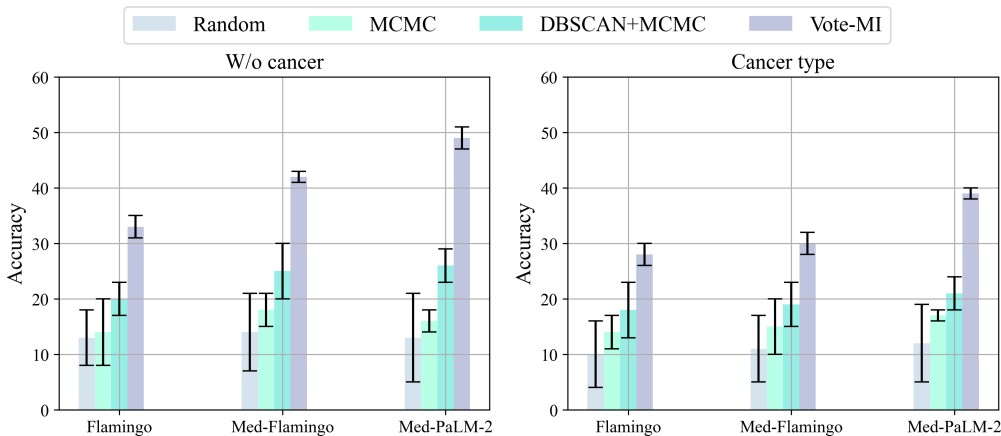

Figure 11: Performance comparisons of various VLMs under two disease diagnostics downstream task settings during ablation studies settings, including Random, MCMC, DBSCAN+MCMC, and Vote-MI, show that the final Vote-MI representative selection method consistently improves performance by a large margin in zero-shot learning with pre-trained vision-language models. These findings highlight the effectiveness of each component of the Vote-MI method and its overall superiority over the baseline methods

Table 11: In-context learning results with randomly-selected, radiologist-selected, and Vote-MI representative selection methods on the BrainMD dataset, with a selection budget of 2, 4, or 8. Across the board, representative selection with Vote-MI substantially outperforms the randomly selected baseline for few-shot learning.

| Size | Method | Tasks | | |
|------|--------|-------|---|---|
| $\mathcal{L}$ | Selection | W/o cancer | Cancer types | Cancer status |
| 2 | Random | 21.3 ($\pm$ 5.6) | 18.3 ($\pm$ 7.1) | 19.6 ($\pm$ 4.0) |
| 2 | MCMC | 23.4 ($\pm$ 3.2) | 20.9 ($\pm$ 4.8) | 21.3 ($\pm$ 3.6) |
| 2 | DBSCAN+MCMC | 29.8 ($\pm$ 3.3) | 26.1 ($\pm$ 4.7) | 27.9 ($\pm$ 3.5) |
| 2 | Vote-MI | **43.5 ($\pm$ 2.3)** | **35.3 ($\pm$ 4.5)** | **37.7 ($\pm$ 3.8)** |
| 4 | Random | 28.1 ($\pm$ 5.1) | 24.5 ($\pm$ 6.2) | 26.7 ($\pm$ 5.8) |
| 4 | MCMC | 30.9 ($\pm$ 3.4) | 26.2 ($\pm$ 4.3) | 27.6 ($\pm$ 4.0) |
| 4 | DBSCAN+MCMC | 36.8 ($\pm$ 3.5) | 31.1 ($\pm$ 4.2) | 32.8 ($\pm$ 4.1) |
| 4 | Vote-MI | **50.7 ($\pm$ 2.8)** | **43.0 ($\pm$ 3.9)** | **42.7 ($\pm$ 4.0)** |
| 8 | Random | 30.1 ($\pm$ 6.1) | 28.7 ($\pm$ 6.5) | 30.4 ($\pm$ 6.3) |
| 8 | MCMC | 34.2 ($\pm$ 3.3) | 32.7 ($\pm$ 4.5) | 34.8 ($\pm$ 4.2) |
| 8 | DBSCAN+MCMC | 39.8 ($\pm$ 3.2) | 37.5 ($\pm$ 4.4) | 36.6 ($\pm$ 4.3) |
| 8 | Vote-MI | **55.3 ($\pm$ 3.1)** | **46.0 ($\pm$ 4.2)** | **46.7 ($\pm$ 4.1)** |