# OpenReview forum: "Enhancing vision-language models for medical imaging: bridging the 3D gap with innovative slice selection"
_NeurIPS.cc/2024/Datasets_and_Benchmarks_Track — NeurIPS 2024 Track Datasets and Benchmarks Poster_

### Official Review · Reviewer_7Ndp · 2024-07-14
**N/a**

**Rating:** 5
**Confidence:** 4
**Correctness:** The dataset is designed in a right way.
**Clarity:** Clarity is fine.

**Review:**

In this study, the authors introduce BrainMD, the first-ever large multimodal longitudinal brain tumor dataset named BrainMD. The dataset comprises 2453 studies from 561 patients, each including 3D medical images, radiology reports, and longitudinal medical records. A novel representative slice selection method called Vote-MI is proposed to enable large volumetric machine learning models (VLMs) to process 3D imaging data efficiently by capturing crucial information from volumetric data. The authors also establish two benchmarks based on the BrainMD dataset for diagnosing and prognosticating outcomes of brain tumors.

**Strengths:**

(1)	The primary strength lies in the construction of a multi-modal dataset, this study pioneers the field and pushes its boundaries forward.
(2)	In order to facilitate the application of large vision-language models (VLMs) to 3D medical images, a carefully designed slice selection method is introduced to enable efficient inference.
(3)	The authors have made the code and example data publicly available to facilitate better evaluation and reproducibility of the results.

**Additional Feedback:**

NA

**Documentation:**

The dataset and baselines are well-documented and available. However, some details of the dataset are unclear, such as diversity and statistical information.

**Ethics:**

I do not suspect any ethical concerns.

**Limitations:**

The performance of the Visual Question Answering (VQA) task imposes limitations on its application scenarios.

**Opportunities For Improvement:**

(1)	While this work highlights the novelty of combining 2D/3D medical images with both radiology reports and longitudinal medical records, it is important to acknowledge that similar approaches have been proposed in prior studies [1, 2]. It would be beneficial to mention and discuss these relevant works in order to provide a comprehensive overview of the existing research landscape and highlight the unique contributions of the current study.
(2)	The number of comparison methods for slice selection in this study is limited. In the field of video actin recognition, frame selection is a commonly used method. It would be advisable to mention and compare the relevant methods used in video actin recognition, such as those proposed in [3, 4]. By including these comparisons, a more comprehensive evaluation of the proposed slice selection method can be achieved, highlighting its advantages and distinguishing features.
(3)	The downstream Visual Question Answering (VQA) task predominantly focuses on classification, which may limit the broader application of this dataset.

[1]. Bai, Fan, et al. "M3d: Advancing 3d medical image analysis with multi-modal large language models." arXiv preprint arXiv:2404.00578 (2024).
[2]. Hamamci, Ibrahim Ethem, Sezgin Er, and Bjoern Menze. "Ct2rep: Automated radiology report generation for 3d medical imaging." arXiv preprint arXiv:2403.06801 (2024).
[3].  Zhao, Mingjun, et al. "Search-map-search: a frame selection paradigm for action recognition." Proceedings of the IEEE/CVF Conference on Computer Vision and Pattern Recognition. 2023.
[4]. Wu, Zuxuan, et al. "Adaframe: Adaptive frame selection for fast video recognition." Proceedings of the IEEE/CVF Conference on Computer Vision and Pattern Recognition. 2019.

**Relation To Prior Work:**

Differences with previous 3D medical image analysis are not well demonstrated.

**Summary And Contributions:**

In this study, the authors introduce BrainMD, the first-ever large multimodal longitudinal brain tumor dataset named BrainMD. The dataset comprises 2453 studies from 561 patients, each including 3D medical images, radiology reports, and longitudinal medical records. A novel representative slice selection method called Vote-MI is proposed to enable large volumetric machine learning models (VLMs) to process 3D imaging data efficiently by capturing crucial information from volumetric data. The authors also establish two benchmarks based on the BrainMD dataset for diagnosing and prognosticating outcomes of brain tumors.

---

> ### Author Rebuttal · Authors · 2024-08-17
>
> Dear Reviewer 7Ndp,
>
> Thank you for all the comments, which are really helpful. Our responses are as follows:
>
> **More relevant works to provide a comprehensive overview:** Thank you for your comments and for pointing out the overlook in our literature review. As indicated in the **General comments** section above, we have added a new *Section 2.3*, titled "Medical Vision Language Model," under *Section 2 Related Work*. As shown at the end of this rebuttal, this new *Section 2.3* includes citations of the two significant papers you mentioned, *M3D* and *CT2Rep*, along with another recent paper, *Merlin*, to showcase the advancements in 3D vision language models. Additionally, we highlight several 2D vision language models in this section, such as *LLaVA-Med* and *Med-Flamingo*. We hope this addresses your concern on this point.
>
> **Compare slice selection method in video recognition field:** We appreciate your expertise in video action recognition, which we are not familiar with, and thank you for pointing out its connection to our 2D slice selection method. We find the methods you mentioned, such as Search-Map-Search and AdaFrame, particularly promising. Search-Map-Search combines hierarchical search algorithms with supervised learning to select optimal frames from a video, while AdaFrame uses a Long Short-Term Memory network to identify representative frames with global context. However, given the time constraints of the rebuttal phase, conducting a high-quality experimental comparison could be stressful. A comprehensive comparison for example requires evaluating (1) slice selection accuracy, (2) zero-shot downstream task performance with various VLMs, and (3) few-shot downstream task performance with different VLMs. We will address these experiments in our future research and we plan to integrate the approaches you mentioned to enhance our method in the future.
>
> Furthermore, our primary goal in this paper is to provide a comprehensive dataset and a straightforward baseline with reasonable performance. Achieving higher performance levels would require extensive research beyond the scope of this paper and is more suited to future work. We hope you understand our standing point here and hope this addresses your concerns.
>
> Lastly, thank you for your insightful comments on the **broader applications of our dataset**. Your feedback motivates us to explore additional medical applications beyond the two VQA benchmarks presented in the paper. With our dataset comprising 3D medical images, associated radiology reports, and longitudinal medical records from 2,453 studies, we believe it offers significant potential for advancing research across various medical image and report-related tasks. Potential uses include fine-tuning vision-language models for brain tumor research, generating reports, and predicting clinical outcomes.
>
> ### New added Section 2.3
> **2.3 Medical Multi-modal Vision Language Model**
> Recent research [32, 57] highlights the effectiveness of multimodal vision-language models (VLMs) in integrating image and text data for a variety of tasks. These models combine the perceptual capabilities of vision models [39, 46] with the generative power of large language models (LLMs) [42,11,13], gaining significant traction, particularly in medical image analysis. Existing medical VLMs [31, 48, 2, 36] often fine-tune publicly available 2D VLMs on medical image and text data to perform tasks such as image-text retrieval, report generation, and visual question answering. Models like LLaVA-Med [31], Med-PaLM-2 [48], and MedFlamingo [36] are derived from LLaVA [34], PaLM-E [12], and Flamingo [2], respectively.
>
> However, these methods face challenges when applied to 3D medical images, such as CT and MRI scans, which contain rich spatial information. The common approach of slice-by-slice analysis is computationally expensive and often inadequate. While models like RadFM [52] support both 2D and 3D images, they primarily focus on text generation tasks like visual question answering (VQA) and generally underperform. More specialized VLMs, such as M3D-LaMed [5], Ct2rep [18], and Merlin [8], are designed specifically for 3D medical image analysis, tackling tasks like report generation and VQA, and pioneering vision question-answering tasks. Despite these advancements, 3D VLMs continue to struggle due to the lack of large, paired 3D image-report datasets and the high computational demands of model training.

---

### Official Review · Reviewer_Y3Xm · 2024-07-21

**Rating:** 8
**Confidence:** 4
**Correctness:** Yes
**Clarity:** Yes

**Review:**

Collecting and open-sourcing 3D medical imaging data that is paired with radiology reports, etc. is important for advancing the field further, as data is one of the most crucial obstacles for building a good VLM in the medical domain, and it is definitely the largest one that makes the gap large against VLMs that deal with natural images. In this regard, the release of BrainMD along with its annotated counterparts is important and valuable in and of its own.

Vote-MI is interesting, and this goes orthogonal to the current trend of 3D VLMs for medical imaging, where one tries to find a way to incorporate all the information from all the slices.

However, the related works section is poorly written, and the main section explaining about the vote-MI is without much motivation. There is little detail on how the datasets were constructed. Overall, the paper is poorly written.

**Strengths:**

1. Collecting and open-sourcing 3D medical imaging data that is paired with radiology reports, etc. is important for advancing the field further, as data is one of the most crucial obstacles for building a good VLM in the medical domain, and it is definitely the largest one that makes the gap large against VLMs that deal with natural images. In this regard, the release of BrainMD along with its annotated counterparts is important and valuable in and of its own.

2. Vote-MI is interesting, and this goes orthogonal to the current trend of 3D VLMs for medical imaging, where one tries to find a way to incorporate all the information from all the slices.

**Additional Feedback:**

-

**Documentation:**

Yes

**Limitations:**

Yes

**Opportunities For Improvement:**

1. The related works section covers zero-shot learning, few-shot learning, and multi-modal dataset. This is too broad and not too much relevant from the topic of the paper. I believe the authors should focus more on "medical" multi-modal dataset, and the current established methods for dealing with 3D medical data.

2. Section 3 focuses mostly on the sequential algorithmic steps that Vote-MI takes, but much less on why each design components were chosen, and why these were chosen rather than other possible choices. Please clarify.

3. There are 4 contributions of the paper, but the paper focuses very much only on Vote-MI. The dataset is equally interesting, if not more. The main paper should focus more on the design of the dataset, how the annotations were made and filtered, etc.

4. Table 3: when evaluating the performance of Vote-MI, the comparison should also be done against using GT (annotated by radiologists), and simply taking the center slices from all three axes.

Also, the hyperlinks are not working properly.

**Relation To Prior Work:**

More works should be done on strengthening the related works section.

**Summary And Contributions:**

There are several contributions in the paper.

1. **BrainMD**: a dataset is released, encompassing 7 different types of brain tumors, 2k annotated 3D MRI brain scans paired with textual data such as radiology reports, medical records, and demographic info.

2. **BrainMD-select**: An additional annotation provided by board-certified radiologists, marking of which (axial, sagittal, coronal) slice is the most important in the 3D scans, is proposed.

3. **BrainBench**: BrianMD coupled with text-related tasks such as VQA and disease diagnosis, is proposed.

4. **Vote-MI**: an unsupervised method that selects the most important 2D slice from 3D volume, is proposed.

---

> ### Author Rebuttal · Authors · 2024-08-17
>
> Dear Reviewer Y3Xm,
>
> Thank you very much for the kind reviews and your support for acceptance. It is our pleasure to respond to your comments.
>
> **Section 3 focuses ......, but much less on why each design component was chosen, and why these were chosen rather than other possible choices:** We agree with the reviewer's comments on Section 3 and fully agree that it lacks explanations on the design choices of each component. In fact, we have taken serious consideration about algorithmic design and the use of each component. The corresponding ablation study of each component is shown in *Appendix Section D*. These findings from ablation studies highlight the effectiveness of each component of the Vote-MI method and its overall superiority over the baseline methods. Specifically, MCMC outperforms random sampling by exploring the sample space more efficiently and thoroughly, targeting relevant areas more frequently. DBSCAN generally surpasses by identifying clusters of arbitrary shapes, not requiring pre-specified cluster numbers, and handling noise and outliers better, leading to more robust clustering results.
>
> We also take your comments about **comparison between GT (annotated by radiologists) and the center slices from all three axes** into full consideration. To address your comments, we have conducted even more experiments to verify the performance using center slices. The results are shown in the **attached PDF**. Overall, compared to random selection, there is limited accuracy improvement from center slice selection. However, center slice selection performs worse than the Vote-MI method and even further worse than the radiologist selection. As shown in the attached PDF file, Figure 1 shows our results from zero-shot learning over the BrainMD dataset. Table 1 shows our results from few-shot learning over the BrainMD dataset.
>
> **The hyperlinks are not working properly:** We have made sure all the hyperlinks are working now. The hyperlink to the dataset and code is attached here: https://anonymous.4open.science/r/BrainMD-16CC/README.md
>
> Last but not least, we appreciate your points on **related works section writing** and **more space on dataset and annotations**. We revised our paper accordingly, as outlined in the **General comments** section above. We hope these revisions address the reviewer's concerns. For example:
>
> - (1) We revised *Section 2* (Related Work) by merging *Section 2.1* (Zero-shot learning) and *Section 2.2* (Few-shot learning).
> - (2) We added a new *Section 2.3* to cover current methods for handling 2D/3D multi-modal medical data. The revised version of *Section 2* (Related Work) is shown as follows.
> - We moved the technical details of *Section 3* (Vote-MI) to the appendix. We also integrated the content of *Appendix C* into the main paper, emphasizing dataset design and annotation, statistical distribution, and benchmark definition and distribution.
>
>
>
>
>
>
> ### Revised Section 2.1
> **2.1 Zero-shot and Few-shot Learning**
> Zero-shot learning (ZSL) [54] enables models to predict classes they haven’t been explicitly trained on by leveraging auxiliary information. This approach addresses the challenge of acquiring labeled data for every class, especially in domains like rare medical diseases. Various ZSL methodologies include attribute-based [23], embedding-based [56], and generative approaches [29]. The versatility of ZSL has significant implications in vision and language processing, enhancing models’ ability to generalize across diverse and unseen categories.
>
> Few-shot learning [50] requires only a few annotated examples per test instance, avoiding the need for extensive fine-tuning. Recent research has proposed strategies to enhance few-shot learning, such as meta-training [33, 41], task instructions [25], and task formulation [21]. The selection of few-shot examples is crucial, with studies questioning the necessity of correct labels. Our work introduces representative slice selection within a few-shot learning framework, emphasizing the importance of representative examples on the performance of VLMs.
>
> ### New added Section 2.3
>
> **2.3 Medical Multi-modal Vision Language Model**
>
> Recent research [32, 57] highlights the effectiveness of multimodal vision-language models (VLMs) in integrating image and text data for a variety of tasks. These models combine the perceptual capabilities of vision models [39, 46] with the generative power of large language models (LLMs) [42,11,13], gaining significant traction, particularly in medical image analysis. Existing medical VLMs [31, 48, 2, 36] often fine-tune publicly available 2D VLMs on medical image and text data to perform tasks such as image-text retrieval, report generation, and visual question answering. Models like LLaVA-Med [31], Med-PaLM-2 [48], and MedFlamingo [36] are derived from LLaVA [34], PaLM-E [12], and Flamingo [2], respectively.
>
> However, these methods face challenges when applied to 3D medical images, such as CT and MRI scans, which contain rich spatial information. The common approach of slice-by-slice analysis is computationally expensive and often inadequate. While models like RadFM [52] support both 2D and 3D images, they primarily focus on text generation tasks like visual question answering (VQA) and generally underperform. More specialized VLMs, such as M3D-LaMed [5], Ct2rep [18], and Merlin [8], are designed specifically for 3D medical image analysis, tackling tasks like report generation and VQA, and pioneering vision question-answering tasks. Despite these advancements, 3D VLMs continue to struggle due to the lack of large, paired 3D image-report datasets and the high computational demands of model training.

---

> > ### Comment · Reviewer_Y3Xm · 2024-08-17
> >
> > I thank the authors for thoroughly addressing my concerns. While I was leaning towards acceptance in the first place, I am more convinced that the paper should be accepted, and thus increased my score to 8 accordingly.

---

### Official Review · Reviewer_3dm4 · 2024-07-25
**Useful dataset to enable vision-language medical applications**

**Rating:** 6
**Confidence:** 4

**Review:**

Quality: the dataset consists of MRI scans from 561 patients. This is a subset of a larger dataset from which only the scans with sufficient quality and medical records were selected. Radiology and longitudinal reports accompany each scan. There is also metadata about patients; however, the dataset is de-identified. Thus, I think this is a high-quality dataset as compared to many other similar datasets already available.

Clarity: there is a clear description of how the dataset was obtained and processed.

Originality and significance: perhaps not a very original idea, nevertheless this is a significant contribution that puts together 3D MRI scans with radiology reports and longitudinal studies.

The paper spends a considerable amount of space describing a method for representative scan selection and the two benchmarks. I think these are not that essential for introducing the dataset and perhaps more space should be dedicated to describing the dataset rather than these additional auxiliary models.

**Strengths:**

The provided dataset is significant enough to complement the existing brain MRI datasets. It could be very instrumental in both more traditional applications, such as tumor segmentation, as well as emerging applications based on LLMs and vision. The dataset includes human-generated radiology reports, which are presumably high-quality. It also includes longitudinal studies which are important to provide annotations for disease progression. Gender, age, and racial data are also provided with the scans. All this makes the dataset quite comprehensive.

**Additional Feedback:**

Nothing in particular.

**Clarity:**

The paper clarity can be improved by spending more space on the dataset description vs. benchmarks, etc.

**Correctness:**

The dataset seems to be constructed in a sensible way by reviewing and eliminating incomplete MRI records.

**Documentation:**

There seems to be a sufficient documentation about the dataset, how it was collected, and what it is intended for.

**Ethics:**

I do not think there are any ethical concerns; the authors obtained IRB approval from their institution for the relevant records.

**Limitations:**

The authors have discussed several limitations of the presented dataset. I do not see a reason for a negative societal impact of this work. For any dataset of this nature, it is very difficult to collect more than just a few hundreds of medical cases and it is very likely that these cases will be biased towards a specific population, as the case of this dataset. However, this bias does not diminish the significance of this dataset compared to what is already available.

**Opportunities For Improvement:**

The writeup spends a lot of space on the method for representative slice selection, which apparently does not work that well when compared to a radiologist's choice, as well as the benchmarks that are not that essential for presenting the dataset.

**Relation To Prior Work:**

There is a clear discussion about prior datasets that are used in the relevant community.

**Summary And Contributions:**

The submission presents a large-scale medical dataset called BrainMD with multiple modalities, comprising of health records for 2,453 high-quality MRIs from 561 patients, complete with radiology reports, and structured data from medical records.  The paper also discusses two benchmarks and a method for representative slice selection. This could be very useful data, particularly if one needs to train an LLM to generate clinical reports.

---

> ### Author Rebuttal · Authors · 2024-08-17
>
> Dear Reviewer 3dm4,
>
> We appreciate your insightful comments and the time you invested in reviewing our manuscript. Below, we address each of your points of concern.
>
> We appreciate the reviewer's feedback on **the need for additional details on the dataset and benchmarks** and totally agree that the center of this paper should be focused on the dataset and benchmarks. To address this, we revised the paper as outlined in the above **General Comments** section. Specifically, we moved the technical details of *Section 3*, which covers the Vote-MI slice selection methods, to the appendix. This allows us to enhance the main paper by incorporating the contents of *Appendix C*, with a focus on the design and annotation of the dataset, its statistical distribution, and the benchmarks' definition and distribution. This adjustment provides more space to thoroughly describe the dataset.
>
> **Slice selection method does not work well compared to radiologist:** We appreciate the reviewer's feedback and apologize for any confusion caused. We acknowledge that, at first glance, Vote-MI's performance might not seem to surpass that of radiologist selection. However, it is important to consider that radiologist selection involves significant effort and specialized domain knowledge, with over five certified radiologists dedicating more than 40 hours each. Our proposed method eliminates the need for manual human labeling and offers a more straightforward, single-pass slices selection approach. Despite its simplicity, it effectively enhances the application of large vision-language models to 3D medical images, leading to substantial improvements in downstream task performance. We believe this method only represents a starting point for the proposed dataset and may inspire future advancements in this area.
>
> We appreciate the reviewer's insightful comments regarding the **benchmarks that may not be essential for presenting the dataset**. This feedback encourages us to consider broader medical applications beyond the two VQA benchmarks currently discussed in the paper. Given that our dataset includes 3D medical images, corresponding radiology reports, and longitudinal medical records from 2,453 studies, we believe it serves as a valuable resource for advancing research in various medical image and report-related tasks. Potential applications include fine-tuning vision-language models for brain tumor analysis, generating radiology reports, and predicting clinical outcomes.

---

> > ### Comment · Reviewer_3dm4 · 2024-08-19
> >
> > Thank you for addressing these concerns. I appreciate your willingness to restructure the manuscript to bring the focus on the dataset itself, this helps to make the contribution more clear. I updated my initial assessment.

---

### Author Rebuttal · Authors · 2024-08-17

## General comments

Dear reviewing committee,

We thank the reviewers for their thoughtful feedback and constructive criticisms of our manuscript. We want to emphasize that our primary contribution is the uniqueness and completeness of our dataset, which contains MRI images, radiology reports, longitudinal EHRs, and representative 2D slices of 3D images from over 2,453 studies and 561 patients. We are thankful that all reviewers have recognized this strength. For example, as per reviewers' comments, the dataset and benchmarks delivered in this paper are *"very useful data, particularly to train an LLM to generate clinical reports"* (**Reviewer 3dm4**), *"definitely the largest one that makes the gap large against VLMs"* (**Reviewer Y3Xm**), *"pioneer the field and pushes its boundaries forward"* (**Reviewer 7Ndp**). Furthermore, we contribute two benchmarks (two diagnosis tasks and one prognosis task), and a tailored 2D slices selection baseline method, which initiates using our proposed dataset and setup using the state-of-the-art 2D vision language models.

The reviewers have provided valuable suggestions regarding related works, writing, and experiments. We appreciate these insights and have made the following revisions based on their feedback (further detailed responses to each reviewer's comments are provided in the following sections):

- We reorganized the paper and improved the writing on the dataset, following the guidance of **Reviewer 3dm4** and **Reviewer Y3Xm**. For example:
  - We moved the technical details of *Section 3: Vote-MI method* to the appendix.
  - We integrated Appendices C (Cohort definition and Dataset documentation) into the main paper, with a major emphasis on dataset design/annotation, dataset statistics, and benchmark label definition and distribution.
  - We revised *Section 2 Related Work* by merging *Section 2.1 Zero-shot learning* and *Section 2.2 Few-shot learning*, and added a new *Section 2.3* to cover current methods for handling multi-modal medical data.

- We discussed relevant papers pointed out by **Reviewer 7Ndp** and the references therein.
- We included additional experiments or explanations in the main paper or appendix:
  - We explained the rationale for choosing each component of the Vote-MI method (**Reviewer Y3Xm**).
  - We conducted experiments comparing ground truth (annotated by radiologists) with center slices from all three axes (**Reviewer Y3Xm**).

More importantly, **Reviewer 7Ndp** has provided several valuable insights regarding future directions for our work. Notably, **Reviewer 7Ndp** highlighted a fascinating connection to video action recognition, which introduces a novel perspective on 2D representation selection within 3D medical images. This connection opens up the possibility of applying advanced techniques from the video action recognition field to enhance our proposed methods. Additionally, **Reviewer 7Ndp** suggested that our dataset could be applied more broadly, such as in fine-tuning vision-language models, generating clinical reports, and predicting patient outcomes. Due to constraints in time and resources, we focused on exploring the two Visual Question Answering (VQA) benchmarks proposed in this paper. However, we anticipate that, with the publication of this paper and dataset, the research community will pursue further applications that expand upon our work.

We are happy to receive high-quality reviews this time; thank you all. Please feel free to communicate with us during the discussion phase if you have any further concerns.

---

> ### Author Response · Authors · 2024-08-30
>
> We are grateful to all the reviewers for their active interaction during the discussion phase, which was fruitful and deepened our understanding of the proposed method and further improved the structure of our paper.
>
> We are encouraged as **Reviewer Y3Xm** and **Reviewer 3dm4** have increased their scores. With their positive feedback, we are confident that we could deliver a stronger revision soon.
>
> On the other hand, since **Reviewer 7Ndp** has not responded to our rebuttal, we would like to reiterate our reply below to gain his or her attention:
>
> - Thank you for your comments and for pointing out the overlook in our literature review. We have added a new Section 2.3, titled "Medical Vision Language Model," under Section 2 Related Work. As shown in the rebuttal, this new Section 2.3 includes citations of the two significant papers you mentioned, M3D and CT2Rep, along with another recent paper, Merlin, to showcase the advancements in 3D vision language models. Additionally, we highlight several 2D vision language models in this section, such as LLaVA-Med and Med-Flamingo. We hope this addresses your concern on this point.
>
> - We appreciate your expertise in video action recognition and thank you for pointing out its connection to our 2D slice selection method. We find the methods you mentioned, such as Search-Map-Search and AdaFrame, particularly promising. Search-Map-Search combines hierarchical search algorithms with supervised learning to select optimal frames from a video, while AdaFrame uses a Long Short-Term Memory network to identify representative frames with global context. A comprehensive comparison, for example, requires evaluating (1) slice selection accuracy, (2) zero-shot downstream task performance with various VLMs, and (3) few-shot downstream task performance with different VLMs. We will address these experiments in our future research, and we plan to integrate the approaches you mentioned to enhance our method in the future.
>
> - Thank you for your insightful comments on the broader applications of our dataset. Your feedback motivates us to explore additional medical applications beyond the two VQA benchmarks presented in the paper. With our dataset comprising 3D medical images, associated radiology reports, and longitudinal medical records from 2,453 studies, we believe it offers significant potential for advancing research across various medical image and report-related tasks. Potential uses include fine-tuning vision-language models for brain tumor research, generating reports, and predicting clinical outcomes.
>
> We hope this will further reduce **Reviewer 7Ndp's** concerns so that all reviewers can reach a better and more solid consensus. Thank you all once again for engaging with us and providing invaluable feedback. Please feel free to reach out with any additional concerns or questions before the discussion phase ends.

---

### Decision · Program_Chairs · 2024-09-26

**Decision:**

Accept (Poster)

**Comment:**

The paper introduces BrainMD, a significant multimodal brain tumor dataset, along with Vote-MI, a method for selecting representative slices from 3D medical imaging data. The dataset is of high quality and paired with radiology reports and EHRs, and Vote-MI shows clear performance improvements in zero-shot and few-shot learning tasks. While the related work section could better situate the contribution within existing research, and further explanation of the design rationale for Vote-MI is needed, the authors have adequately addressed most reviewer concerns through revisions. The dataset's release and the method’s applicability to vision-language models enhance the paper's impact on the field.